# Throwaway Shadows Using Parallel Encoders Generative Adversarial Network

**Kamran Javed** [1,*], **Nizam Ud Din** [2] , **Ghulam Hussain** [2] **and Tahir Farooq** [3]

1  National Centre of Artificial Intelligence (NCAI), Saudi Data and Artificial Intelligence Authority (SDAIA), Riyadh 11543, Saudi Arabia
2  College of Information and Communication Engineering, Sungkyunkwan University, Suwon 16419, Korea; nizam@skku.edu (N.U.D.); hussain@skku.edu (G.H.)
3  Leverify LLC, 16301 NE 8th Street Suite 206, Bellevue, WA 98008, USA; Tahir@leverify.com
*  Correspondence: kamran.javed@giki.edu.pk

**Abstract:** Face photographs taken on a bright sunny day or in floodlight contain unnecessary shadows of objects on the face. Most previous works deal with removing shadow from scene images and struggle with doing so for facial images. Faces have a complex semantic structure, due to which shadow removal is challenging. The aim of this research is to remove the shadow of an object in facial images. We propose a novel generative adversarial network (GAN) based image-to-image translation approach for shadow removal in face images. The first stage of our model automatically produces a binary segmentation mask for the shadow region. Then, the second stage, which is a GAN-based network, removes the object shadow and synthesizes the effected region. The generator network of our GAN has two parallel encoders—one is standard convolution path and the other is a partial convolution. We find that this combination in the generator results not only in learning an incorporated semantic structure but also in disentangling visual discrepancies problems under the shadow area. In addition to GAN loss, we exploit low level L1, structural level SSIM and perceptual loss from a pre-trained loss network for better texture and perceptual quality, respectively. Since there is no paired dataset for the shadow removal problem, we created a synthetic shadow dataset for training our network in a supervised manner. The proposed approach effectively removes shadows from real and synthetic test samples, while retaining complex facial semantics. Experimental evaluations consistently show the advantages of the proposed method over several representative state-of-the-art approaches.

**Keywords:** shadow removal; image restoration; image reconstruction; partial convolution

## 1. Introduction

Facial images have become one of the most popular sources of images captured daily, transmitted through electronic media and/or shared on the social networks. In the real world, these images are often corrupted by some image conditions, especially the shadows of different objects. This not only degrades image quality but also affects the visual appearance of the image. The main objective of this research is to automatically detect and remove the shadow of an object from the facial images and produce a shadow free image. Most of the previous shadow removal works deal with removing shadow from the scene images and to the best of our knowledge there is no previous work for shadow removal from the facial images. Since faces have a complex semantic structure, shadow removal from facial images is an extremely challenging problem in computer vision.

Traditional shadow removal works [1,2] by using a physical model. These non-trivial methods take lot of processing time and suffer for shadow removal in facial images. On the other-hand, learning-based methods [3–6] outperformed non-learning based methods for the shadow removal task. Although they produce good results as compared to the

traditional algorithms for removing shadow from scene images, they are unable to remove shadows from facial images due to the complex nature of face semantics.

In this work, instead of improving or modifying the previous deep learning based shadow removal model for removing shadow from face images, we took a totally different approach by using an image inpainting approach as a shadow removal model. Image inpainting is the method used to reconstruct lost or damaged parts of an image. The current state-of-the-art deep learning based image inpainting methods [7–11] reconstruct the damaged region by providing the mask of the damaged part. Refs. [7,8] fill the missing pixel by copying similar patches from the surrounding region. Ref. [10] used some guidance information to reconstruct the corrupt part of the image while [12] use two discriminators to enforce global coherency. Some models [9,11,13] use two-stage networks by generating coarser results in the first stage and refine it in the second stage.

All of the above mentioned deep learning works only use standard convolution as the backbone operation of a neural network. The standard convolution employs the same filter weights all over the image, nevertheless pixels are valid or affected. Eventually, it generates a well-incorporated structure under the affected area but fails to remove visual artifacts, particularly at the boundary of the affected and valid area as mentioned in [14,15]. In the shadow removal problem, this issue becomes more severe because most of the time the shadow area is large and an irregular shape. To overcome these issues, many researchers use extensive post processing steps and/or additional refinement stages as in [8,11]. To incorporate with irregular shape recovery and limitations of standard convolution, improved convolution—called partial convolution [14]—is proposed.

In partial convolution, convolution is only employed on a valid pixel area and is re-normalized. A segmentation mask is used to locate a valid pixel area [14]. The valid pixel mask is updated after each iteration to compute new valid pixels. Additionally, our approach does not have any post processing or refinement stages. In this paper, we are considering the shadow part of an image as a damaged or corrupted area. The first segmentation mask of a shadow is generated by a simple convolution auto-encoder network and then use that mask of the shadow part along with input image, to reconstruct the shadow part of the image. We propose a GAN based deep network that takes an input image along with the binary mask of the shadow region and produces a shadow free image that is consistent both visually and structurally. The main contributions of this work are summarized as follows:

- We propose a novel GAN-based image inpainting approach to remove the shadows of objects from facial images;
- Our method generates a well-incorporated semantic structure and disentangles the visual discrepancies issue under the shadow region by employing a combined parallel operation of standard and partial convolution in a single generator model;
- To train our shadow removal network in a supervised manner, we create a paired synthetic shadow dataset using facial images from the CelebA dataset;
- Our model removes the shadow and creates perceptually better outputs with fine details in challenging facial images.

The remaining parts of the paper are organized as follows. Section 2 covers related works. The architecture of the shadow removal network is described in Section 3. Section 4 covers the experimental setting. Section 5 details the results and discussion.

## 2. Related Work

**Generative Adversarial Network (GAN):** GANs have shown a promising ability for image generation problems [16]. GAN is a two network model; one is a generator network and other is a discriminator network. The purpose of the generator network is to learn a given data distribution, where the intention of the discriminator network is to estimate the probability that a given sample is real or fake, that is, generated from generator network. GAN uses adversarial training, where the generator and the discriminator networks train alternatively. One popular improvement is to use multiple stages of GAN. Zhang et al.

proposed Stacked Generative Adversarial Networks (StackGAN) [17]. It is a two-stage GAN network to produce a high resolution output from the text description. The first stage GAN network generates low resolution results from the given text description and the output image of Stage-I GAN along with the input text. Then the second stage is fed with the results of the first stage and produce high resolution photo realistic images with fine details. GAN has a proven powerful solution to generate natural looking results [18]. Due to its success for various tasks, GANs are widely used for problems such as domain translation [19–21], texture synthesis [22,23], image inpainting [7,8,11,12,21,24–26] and shadow removal [6,27–30].

**Image inpainting:** The goal of inpainting is to recover the missing part in an image. There are countless applications of inpainting, from removing undesired objects and restoring corrupted regions to adding specific objects. Traditional inpainting approaches propagate the information to fill-in the corrupted area from neighboring pixels [31,32]. All of these methods can only fill-in a small area with stationary texture and failed to inpaint areas where texture and color variance is large. To overcome the texture issue, patch based methods are introduced, which copy similar patches from the input image and paste it into the target image [33,34]. However, this approach works well for non-stationary texture inpainting. Patch based methods have a computational cost because they search in an iterative manner, which is inefficient for real-time implementation.

A pioneer deep-learning based image inpainting method is proposed in [7], which can inpaint the large missing region conditioned on its neighbouring information. The combination of pixel-wise loss and adversarial loss is used for training. However, high frequency details are missing and sometimes generate artifacts in the output images. For better perceptual results, structural inpainting [8], which is based on [7], used perceptual reconstruction loss in addition to existing loss. Structural inpainting can inpaint the complex structures. Khan et al. [13] proposed two stage GAN to de-pixelate the mosaic face image. Their network first removes the mosaic part in the image and then generates face semantics. It works in the coarse-to-fine manner. For better perceptual results, Ref. [35] proposed UMGAN with perceptual loss from the pre-trained network. Refs. [13,35] are limited to square-shape corrupted areas only, but shadows can be an irregular shape. Similar to [7,8,24], we exploit both low-level ($l_1$) loss and high-level (*SSIM*) loss in terms of reconstruction loss to inpaint the region under the shadow.

A two-stage network called EdgeConnect [10] is proposed to inpaint the corrupted image by employing the hallucinated edge information of the corrupted region. Since the results of EdgeConnect rely on the quality of the edge map produced by the edge generator network, so the output suffers when the edge generator network failed to produce a right edge map. New convolution schemes, such as partial convolution [14] and gated convolution [15], were developed to overcome the limitations of the aforementioned methods. These methods produce better results in terms of color correspondence and incorporated semantics.

**Object removal:** The exemplar-based method for texture synthesis is proposed by Criminisi et al. [36]. It inpaints the missing area with plausible texture but fails to generate reasonable results for the regions which do not have similar patches in the image. Improved exemplar based inpainting methods are described in [37] to remove an object from a single image. Normalized cross correlation along with the summation of squared differences is used to find a matching patch in the image. However, it removes the object accurately in simple scenes but the boundary of the removed region has some artifacts. Kamran et al. [9] proposed a two-stage GAN based neural network to remove a microphone object in facial images. It can efficiently remove small objects like a microphone and recover semantics under that but struggle to recover a large area. Recently, Din et al. [38,39] proposed a GAN-based network to effectively remove a large occluded object from facial images.

**Shadow removal:** Shadow removal is one of the popular topics in the computer vision field nowadays, where the goal is to remove shadows from photographs which were taken on a sunny day. Ding et al. [29] proposed a robust attentive recurrent GAN based network

to detect and remove shadows. Their approach is able to remove shadows from complex scene images. The model is very flexible to incorporate sufficient unsupervised shadow images to train a powerful model. As compared to conventional approaches, which uses an illumination model to remove shadows, Ref. [27] proposed a deep neural network, which accurately and automatically estimates the parameters for the model from a single image.

Mask-Shadow GAN, presented in [30] uses un-paired images to remove shadows from scene images. Instead of shadow-free to shadow translation, Mask-Shadow GAN is a deterministic image translation technique, which uses shadow masks as a guidance, which are automatically learned from the real world images with shadow. RIS-GAN is proposed in [28], and exploits the residual and illumination. They explored the correlation between residual, illumination and the shadow by using a unified end-to-end framework. One recent work [6] proposed a method, which first aggregates with context using an aggregation model and then hierarchically aggregates the attentions and features. A shadow matting generative network is trained to generate the shadow images from the corresponding shadow-free images and masks. The shadow matting generative network not only enlarges the scenes in the shadow database but also reduces the color discrepancies.

## 3. Our Method

This section describes the network architecture of the proposed shadow removal method and the details of the objective function we used for training. Our network consists of two stages; in the first stage, we used a convolution auto-encoder to detect the shadow of an object. In the second stage, we used a GAN based image-to-image translation method, which effectively removes shadows in facial images and produces fine details. Figure 1 shows the overall shadow removal architecture.

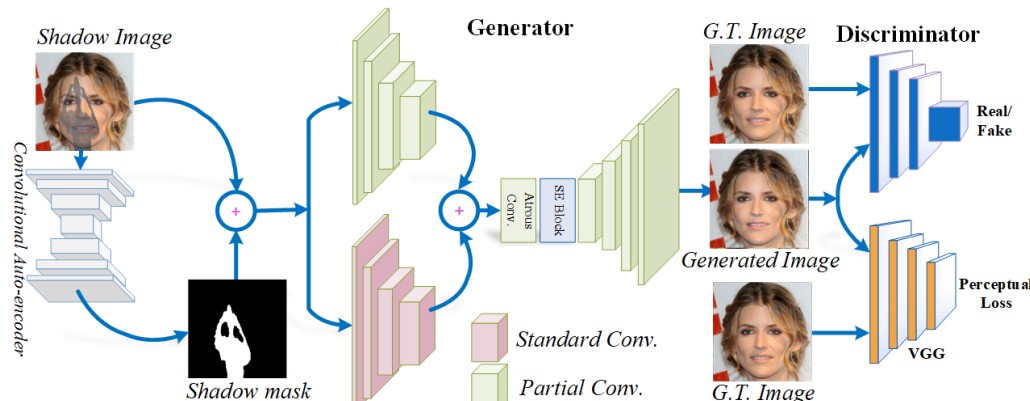

**Figure 1.** Proposed Network Architecture for Shadow Removal.

### 3.1. Network Architecture

The first stage of our network (Convolutional Auto-encoder) consists of a CNN-based encoder and decoder architecture. The encoder consists of five layers where each layer consists of a convolution layer followed by an activation function (*Lrelu*) and an instance normalization layer, except the first layer. The decoder architecture is a mirror copy of the encoder architecture except that convolution is replaced by a deconvolution layer. The convolutional auto-encoder takes the input shadow image and produces a binary segmentation mask for the object's shadow. We used a cross-entropy loss as an objective function between the predicted binary segmentation mask and the corresponding target segmentation map.

Since the second stage (shadow removal) of our network utilizes a GAN based model, it has generator and discriminator networks. The generator network has two parallel encoders; one is a standard convolution path and the other is a partial convolution path. This combination in the generator results not only in learning incorporated semantic structures but also disentangling the visual discrepancies problem under the shadow

area. We start with a UNET-like architecture [40], which has skip connections between the standard convolution encoder and decoder. The purpose of the skip connections is to provide super highways for a gradient during back propagation and avoid the vanishing gradient problem.

Additionally, we used an atrous convolution [41] layer and a squeeze and excitation block [42] between the encoders and the decoder networks of the generator. The intention of atrous convolution is to capture a large field of view for a semantically coherent output and reduce the trainable parameters. To increase the representational power of our architecture, we used a squeeze and excitation block followed by atrous convolution. The purpose of the squeeze and excitation block is to perform dynamic channel-wise feature re-calibration. Moreover, a decoder network is similar to the standard convolution encoder, except transpose convolution is used instead of convolution. Each convolution layer consists of relu+conv+instant norm operation. Our discriminator is a Patch-GAN based architecture, which penalizes the patches instead of each pixel.

### 3.2. Objective Function

To enforce the generator to remove shadow and produce realistic and perceptually correct content under the shadow, we used a joint objective function, which is a combination of four different loss terms. The overall training objective function can be written as follows:

$$\mathcal{L}_{obj} = \alpha.(\mathcal{L}_{l_1} + \mathcal{L}_{ssim}) + \mathcal{L}_{adv} + \beta\mathcal{L}_{perc}, \tag{1}$$

where $\mathcal{L}_{l_1}$ is a pixel level $l_1$ penalty, $\mathcal{L}_{l_{ssim}}$ is a structural penalty, $\mathcal{L}_{adv}$ is a cross entropy adversarial loss and $\mathcal{L}_{perc}$ is a perceptual penalty, which we calculated by measuring the distance between feature map values of the loss network for the generator output and the corresponding ground truth. Particularly, we used a pre-trained VGG-19 [43] as a loss network. The weight of loss terms can be adjusted with respective constants $\alpha$ and $\beta$.

### 4. Experimental Setup

In this section, we present the experimental setting of our shadow removal network. First, we created a synthetic shadow database and then trained our shadow removal network on it. For fair comparison, we retrained state-of-the-art works such as Edge-Connect [10], Partial Convolution [14], Gated Convolution [15] and Ghost-free Shadow removal [6] on a new synthetic database. At evaluation time, we also show results on real world shadow images, collected from the internet.

**Database:** We trained our shadow removal network in a supervised manner. We started with 20,000 randomly selected images from the CelebA Face dataset [44] and created a synthetic shadow database. CelebA face images contain various celebrity images with wild backgrounds and were taken in diverse conditions. We used OpenFace dlib [45] to align the faces using facial landmark positions. This alignment helps the model to generate the face semantics (e.g., eyes) at the right place on the face. Finally, we generate synthetic images by placing the shadow of various objects using Adobe Photoshop. We consider shadows of various objects of different sizes and scales and placed them at various positions in the face image. Corresponding shadow mask images were also created to train the shadow detection stage of our network. Compared to the shadow dataset created by [46,47], we focus on creating shadow images that contain shadows of various objects instead of producing a relit image with hard cast shadows.

**Training setting:** The convolutional auto-encoder network is fed with an input shadow image and generates a binary map of the object's shadow in the input image. The generator of our shadow removal network then takes the pair of input shadow image and its corresponding mask generated by the convolutional auto-encoder and produces an output image without shadow. While the job of discriminator is to differentiate between the generated and ground truth images without shadows. We trained our network with a joint objective function as in Equation (1). The generator network produces an output face image without the shadow. The data split is 70% for training and 30% for testing. There

is no subject overlap between the training and testing sets; both have distinct images. We used the Adam optimizer [48] with learning rate $2 \times 10^{-4}$ and momentum 0.5 to train the shadow removal network. Random crop and random flip techniques were used for data augmentation. We fed 10 images together in a batch. At the start, the discriminator trained quickly and the generator became weak. We first trained the generator network for one hundred epochs and then both the generator network and the discriminate network were trained for five hundred epochs to avoid this problem. We implemented our network on python using the TensorFlow platform [49]. Our training took around three days on an NVIDIA GeForce 1080Ti graphic card. The code with the pre-trained model will be published on GitHub after acceptance of this manuscript.

## 5. Comparison and Discussion

This section presents the quantitative and qualitative compression of our shadow removal method with the state-of-the-art works on both real world shadow images and synthetic shadow images.

### 5.1. Visual Comparison for Facial Images

Figure 2 shows a comparison of our model results with current state-of-the-art representative methods such as EdgeConnect [10], Partial Convolution [14], Gated Convolution [15] and Ghost-free Shadow removal [6]. To make the comparison fair, we train these methods on our synthetic shadow database. Examples in the first couple of rows are the real test images (no ground truth), while the other two rows show the result for the synthetic test sample.

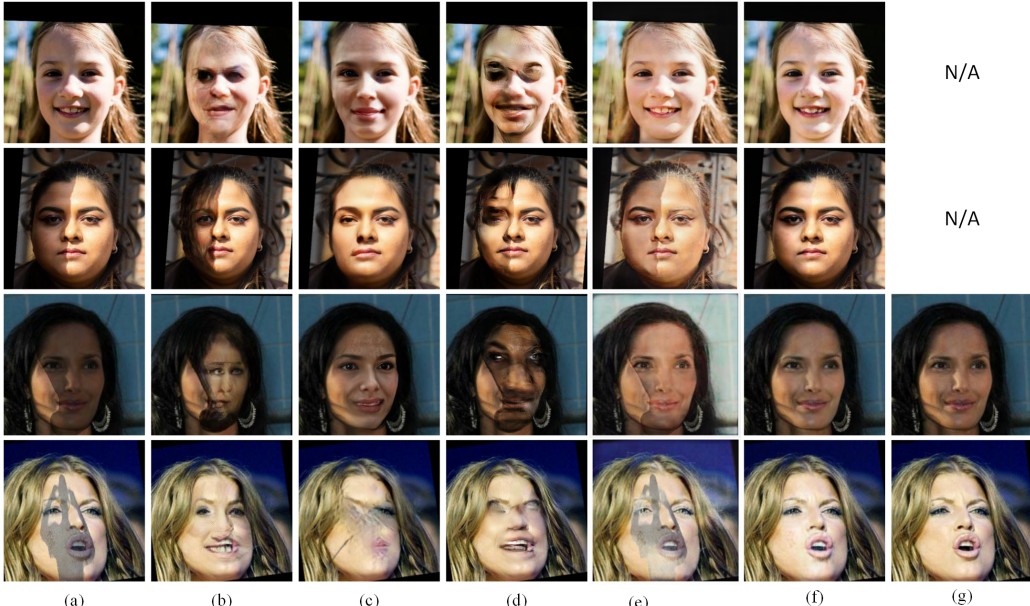

**Figure 2.** Visual comparison of shadow removal. (**a**) Input image, (**b**) EdgeConnect [10], (**c**) Partial Convolution, [14], (**d**) Gated Convolution [15], (**e**) Ghost-free Shadow removal [6], (**f**) Ours, (**g**) Ground truth. **Note:** There is no ground truth for the first couple of rows since these samples are real world shadow images collected from the Internet. The last two samples are from our synthetic database.

As can be seen in Figure 2, our technique plausibly removes shadows from the facial images for both complex real and synthetic test samples. On the other hand, all other representative methods struggle to produce reasonable results. EdgeConnect [10] struggles to produce a proper edge map for a large damaged region of the face resulting in artifacts. Partial Convolution [14] produces sharp results as compared to EdgeConnect and Gated Convolution but still shows artifacts especially at the borders of damaged and undamaged

regions. Ghost-free shadow removal [6] plausibly removes the shadow but is unable to produce natural looking face semantics due to the complex nature of the face semantics.

On the other hand, our model combines the benefits of vanilla and partial convolution encoders. This helps our model in removing the shadow and learning the well-incorporated and artifact-free semantics of the face under the shadow.

Figure 3 shows additional qualitative results of our model for complex and large size shadow samples in our synthetic database. The first column shows the input image, second and third columns show the segmentation map of the object shadow and shadow free image generated by our model, respectively. Last column represents the ground truth for input images. The results show that our model effectively removes different types of complex, large and challenging shadow occlusions from the face images. The last input sample contains shadows created by a lightning effect (not by an occluded object); thus, our model is unable to produce the accurate segmentation mask of the shadow region and a plausible shadow free output.

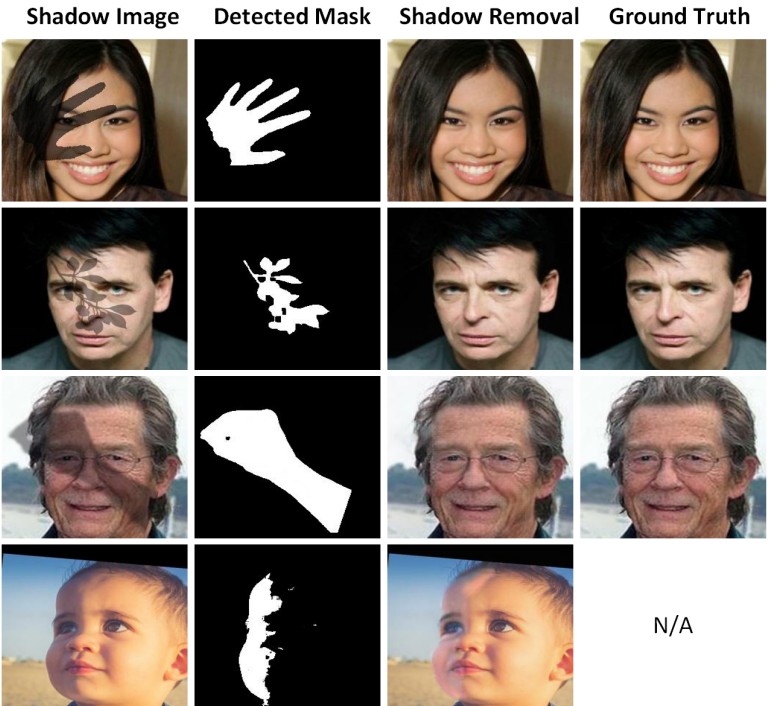

**Figure 3.** Additional qualitative results of our model for complex and large size shadow samples in our synthetic database.

*5.2. Quantitative Evaluation*

In this section, we describe a quantitative comparison of the proposed method with previous state-of-the-art methods such as EdgeConnect [10], Partial convolution [14], Gated convolution [15] and Ghost-free Shadow removal [6] in terms of Root Mean Square Error (RMSE), Naturalness Image Quality Evaluator (NIQE) [50] and Blind Referenceless Image Spatial Quality Evaluator (BRISQUE) [51]. NIQE and BRISQUE measure the naturalness of an image without any reference. Smaller NIQE and BRISQUE scores are better. To measure NIQE and BRISQUE, we used only generated images without providing corresponding ground truths. We have evaluated RMSE on the test images from our synthetic database, which has corresponding ground truths. Table 1 provides a quantitative comparison with previous methods such as EdgeConnect [10], Partial convolution [14], Gated convolution [15] and Ghost-free Shadow removal [6]. The table shows that for the shadow removal problem, the results of our shadow removal method are better than or comparable to those of the state-of-the-art methods.

**Table 1.** Quantitative comparisons of shadow removal in terms of Root Mean Square Error (RMSE), Naturalness Image Quality Evaluator (NIQE), and Blind Referenceless Image Spatial Quality Evaluator (BRISQUE).

| Methods | RMSE | NIQE | BRISQUE |
|---|---|---|---|
| EdgeConnect [10] | 24.73 | 4.429 | 37.01 |
| Partial Conv [14] | 22.41 | 4.248 | 38.60 |
| Gated Conv [15] | 19.00 | 4.614 | **36.44** |
| Ghost-free Shadow removal [6] | 29.44 | 4.190 | 41.30 |
| Ours | **13.91** | **4.005** | 37.93 |

*5.3. Results for Scene Images*

To check the effectiveness of our model for removing shadows in scene images, we trained our proposed model on the publicly available ISTD dataset [4] for removing shadows in scene images. The ISTD dataset consists of 1870 training samples from 135 scenes and 540 test samples from 135 scenes. Our model effectively removed shadow not only in the facial images but also in the scene images as shown in Figure 4. It has potential applications in outdoor photography and surveillance by removing undesired shadows from the images.

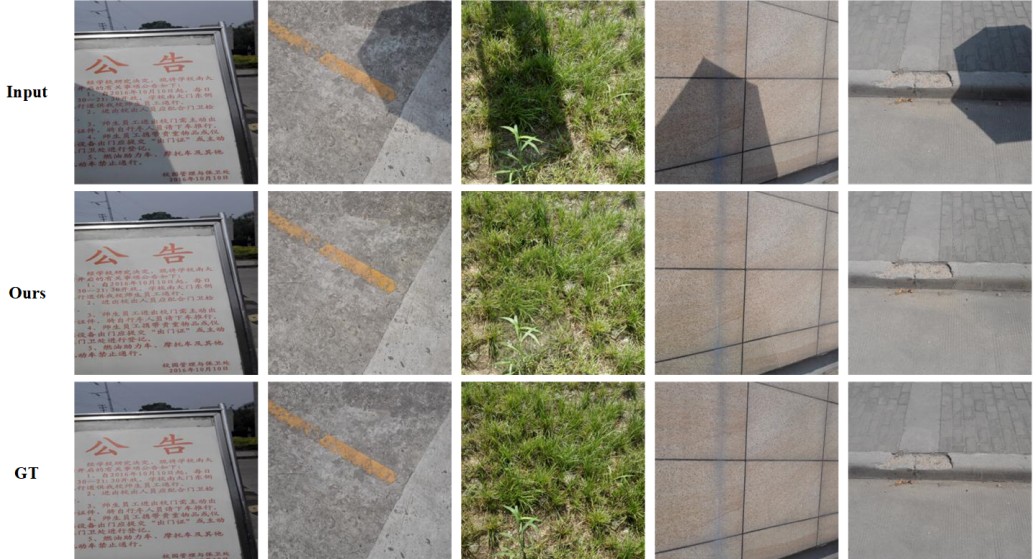

**Figure 4.** Shadow removal results of our proposed method on the scene images from ISTD dataset [4].

## 6. Conclusions

Our shadow removal approach is a GAN based image-to-image translation, which effectively removes shadow in facial images. In this work, we advocate a novel technique for automatically detecting and removing object shadows in facial images. To train our model in a supervised manner, we have created a paired synthetic shadow database. Our method not only generates well-incorporated semantic structures but also disentangles the visual discrepancies problem under the shadow area by employing combined parallel encoders of standard and partial convolution in a single generator model. The performance of our shadow removal method on real world shadow images is adequate although we trained the model using our synthetic shadow database. In the future, we are planning to expend our shadow removal work to automatically detect and remove shadows from lighting effects and occluded objects.

**Author Contributions:** K.J. developed the method; N.U.D. performed the experiments; K.J., N.U.D. did the analysis; and N.U.D., G.H., T.F. and K.J. wrote the paper. K.J., T.F. and N.U.D. proof read the paper. All authors have read and agreed to the published version of the manuscript.

**Funding:** The APC for this research was funded by National Centre of Artificial Intelligence (NCAI) at Saudi Data and Artificial Intelligence Authority (SDAIA), Riyadh, Saudi Arabia.

**Institutional Review Board Statement:** Not applicable.

**Informed Consent Statement:** Not applicable.

**Data Availability Statement:** Not applicable.

**Conflicts of Interest:** The authors declare no conflict of interest.

**Sample Availability:** Not applicable.

## Abbreviations

We used following abbreviations in our manuscript:

| | |
|---|---|
| GAN | Generative Adversarial Networks |
| SSIM | StructuralSIMilarity |
| BRISQUE | Blind/Referenceless Image Spatial Quality Evaluator |
| RMSE | Root Mean Square Error |
| NIQE | Naturalness Image Quality Evaluator |
| SE | Squeeze and Excitation block |

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
