# Peer review of "Throwaway Shadows Using Parallel Encoders Generative Adversarial Network"

_applsci, doi:10.3390/app12020824_

Round 1
Reviewer 1 Report
The paper proposes a GAN-based method for removing shadows from face images. The authors framed the shadow removal as an image inpainting method and used combined encoders of standard and partial convolution on paired images. The authors created a synthetic dataset by inserting shadows using Photoshop in Celeb A dataset. There are several details and experiments missing in the paper.
- The authors compared their method with other image inpainting techniques, but the objective is to perform shadow removal. Why did not the authors compare their method with existing state-of-the-art methods for shadow removal? The authors claimed that exisitng methods struggle with face images, but they claim that this is the first work of shadow removal from face images, so how did authors come to the conclusion that the existing methods will invariably fail. They should have demonstrated that using empirical analysis. They should compare their method with existing shadow removal schemes to show the effectiveness of their approach.
- The authors generated a synthetic dataset by applying unrealistic shadows using image editing tool. This is not how shadows appear in real-world face images see Figure 3(a) top row. There are so many methods for face image relighting which produce realistic shadows by estimating light source direction -- "Towards High Fidelity Face Relighting with Realistic Shadows," CVPR 2021; "Deep single-image portrait relighting," ICCV 2019. The authors should have used these methods to obtain face images with realistic shadows.
- The authors did not specify what is the split between training data and test data in their synthetic dataset. How did they select which images to apply the effects? How did they select which images to represent real-world examples as shown in Fig 2? They mention they collected it from the Internet, how many images, what is the criterion for selection? Were there any subject overlap between training and test set? Many important details are missing in the description.
- The method requires paired images as input, what happens if the shadow mask is not present which is true for most real-world examples. Existing work such as "Shadow Detection and Removal Using GAN", EUSIPCO 2020 and "Image Shadow Removal Using End-to-End Deep Convolutional Neural Networks", Applied Sciences 2019, use unpaired data. How does the proposed method fare with respect to unpaired data-based methods?
- Shadow detection followed by shadow removal is a challenging problem, how will the proposed method handle the challenge of shadow detection?
- The paper contains typos and requires editing: Line 18-- "collected in people daily lives" Line 20-- "effect visual interpretation" Line 31-- "remove shadow form facial images". Singular vs plural discrepancies throughout the paper "Our method combine the.." "our model effectively remove.."
Author Response
Dear Reviewer,
Please refer to the attached file.
Thank you very much

Reviewer 2 Report
The paper presents a novel architecture of GAN to remove the shadow from facial images. The architecture is well described. Indeed, there are two parallel encoders, one with standard convolution and the other with partial convolution. Also, they used skip connections to avoid vanishing gradient problems. The decoder, which reconstructs the image, is composed of atrous convolution and SE blocks. Atrous convolution reduces the number of trainable parameters. The objective function is a combination of 4 losses. The rest of the architecture is also well described. The authors used 3 other architectures from the literature and exploited the CelebA dataset for testing their architecture. Synthetic shadows have been added to images of CelebA dataset.
To summary, the paper is excellent. However, there are 3 things to correct.
1. line 31 : form -> from
line 224: I think there is an extra space before "Table 1"
2. The authors should explain why they need to align the faces with OpenFace dlib. I do not understand this step.
3. Finally, the authors should share their codes for the architecture.
Author Response

(The authors gave the same response as above.)

Round 2
Reviewer 1 Report
The authors have addressed the suggestions/reviews with adequate details in the revised manuscript. There are minor edits as follows:
- Section 4 Line 195-196: "which helps the model to generate which semantic at which place" - this sentence is confusing and needs to be rewritten to clarify what is the author trying to say here.
- Section 4 Line 199: "Compare to...." should be "Compared to" or "In comparison with..."
- Section 5 Line 235-236: "This helps our model in removing the shadow and learning well incorporated and artifacts free semantics of the face under the damaged region." Please clarify this statement as I am not sure what the authors imply by "well-incorporated and artifacts-free semantics", what are examples of such semantics? And what do the authors mean by "damaged" region? Damaged or occluded?
Author Response
Response to the Reviewer 1 Comments
We very much appreciate the time and effort the reviewer has put into the second stage of the revision.
Comment 1
Section 4 Line 195-196: "which helps the model to generate which semantic at which place" - this sentence is confusing and needs to be rewritten to clarify what is the author trying to say here.
Response
We thanks the reviewer for this necessary correction.
Action
We rewrote the sentence to clarify the meaning at Line 196 in updated manuscript.
Comment 2
Section 4 Line 199: "Compare to...." should be "Compared to" or "In comparison with..."
Response
We thanks the reviewer for this necessary correction.
Action
We changed it, now it is at Line 200 in updated manuscript.
Comment 3
Section 5 Line 235-236: "This helps our model in removing the shadow and learning well incorporated and artifacts free semantics of the face under the damaged region." Please clarify this statement as I am not sure what the authors imply by "well-incorporated and artifacts-free semantics", what are examples of such semantics? And what do the authors mean by "damaged" region? Damaged or occluded?
Response
Well incorporated means generated semantic should match with the remaining face semantics for example if one eye is grey and other eye is under shadow i.e., generated by our network must be grey not black.
We used partial convolution and atrous convolution to address this issue by enlarging the field of view to capture neighbouring information as can be seen in Figure 2 row 4, both eyes are consistent.
Artifacts-free semantics means generated area should have smooth face skin colour not as in Figure 2 column (d).
Damage region mean face area which is under the shadow.